# Graph Contrastive Learning with Augmentations

**Yuning You[1*], Tianlong Chen[2*], Yongduo Sui[3], Ting Chen[4], Zhangyang Wang[2], Yang Shen[1]**
[1]Texas A&M University, [2]University of Texas at Austin,
[3]University of Science and Technology of China, [4]Google Research, Brain Team
{yuning.you,yshen}@tamu.edu, {tianlong.chen,atlaswang}@utexas.edu
syd2019@mail.ustc.edu.cn, iamtingchen@google.com

## Abstract

Generalizable, transferrable, and robust representation learning on graph-structured data remains a challenge for current graph neural networks (GNNs). Unlike what has been developed for convolutional neural networks (CNNs) for image data, self-supervised learning and pre-training are less explored for GNNs. In this paper, we propose a graph contrastive learning (GraphCL) framework for learning unsupervised representations of graph data. We first design four types of graph augmentations to incorporate various priors. We then systematically study the impact of various combinations of graph augmentations on multiple datasets, in four different settings: semi-supervised, unsupervised, and transfer learning as well as adversarial attacks. The results show that, even without tuning augmentation extents nor using sophisticated GNN architectures, our GraphCL framework can produce graph representations of similar or better generalizability, transferrability, and robustness compared to state-of-the-art methods. We also investigate the impact of parameterized graph augmentation extents and patterns, and observe further performance gains in preliminary experiments. Our codes are available at: https://github.com/Shen-Lab/GraphCL.

## 1 Introduction

Graph neural networks (GNNs) [1, 2, 3], following a neighborhood aggregation scheme, are increasingly popular for graph-structured data. Numerous variants of GNNs have been proposed to achieve state-of-the-art performances in graph-based tasks, such as node or link classification [1, 2, 4, 5, 6], link prediction [7] and graph classification [8, 3]. Intriguingly, in most scenarios of graph-level tasks, GNNs are trained end-to-end under supervision. For GNNs, there is little exploration (except [9]) of (self-supervised) pre-training, a technique commonly used as a regularizer in training deep architectures that suffer from gradient vanishing/explosion [10, 11]. The reasons behind the intriguing phenomena could be that most studied graph datasets, as shown in [12], are often limited in size and GNNs often have shallow architectures to avoid over-smoothing [13] or "information loss" [14].

We however argue for the necessity of exploring GNN pre-training schemes. Task-specific labels can be extremely scarce for graph datasets (e.g. in biology and chemistry labeling through wet-lab experiments is often resource- and time-intensive) [15, 9], and pre-training can be a promising technique to mitigate the issue, as it does in convolutional neural networks (CNNs) [16, 17, 18]. As to the conjectured reasons for the lack of GNN pre-training: first, real-world graph data can be huge and even benchmark datasets are recently getting larger [12, 19]; second, even for shallow models, pre-training could initialize parameters in a "better" attraction basin around a local minimum associated with better generalization [11]. Therefore, we emphasize the significance of GNN pre-training.

Compared to CNNs for images, there are unique challenges of designing GNN pre-training schemes for graph-structured data. Unlike geometric information in images, rich structured information of

---

[*]Equal contribution.

various contexts exist in graph data [20, 21] as graphs are abstracted representations of raw data with diverse nature (e.g. molecules made of chemically-bonded atoms and networks of socially-interacting people). It is thus difficult to design a GNN pre-training scheme generically beneficial to down-stream tasks. A naïve GNN pre-training scheme for graph-level tasks is to reconstruct the vertex adjacency information (e.g. GAE [22] and GraphSAGE [23] in network embedding). This scheme can be very limited (as seen in [20] and our Sec. 5) because it over-emphasizes proximity that is not always beneficial [20], and could hurt structural information [24]. Therefore, a well designed pre-training framework is needed to capture highly heterogeneous information in graph-structured data.

Recently, in visual representation learning, contrastive learning has renewed a surge of interest [25, 26, 27, 18, 28]. Self-supervision with handcrafted pretext tasks [29, 30, 31, 32] relies on heuristics to design, and thus could limit the generality of the learned representations. In comparison, contrastive learning aims to learn representations by maximizing feature consistency under differently augmented views, that exploit data- or task-specific augmentations [33], to inject the desired feature invariance. If extended to pre-training GCNs, this framework can potentially overcome the aforementioned limitations of proximity-based pre-training methods [22, 23, 34, 35, 36, 37, 38, 39]. However, it is not straightforward to be directly applied outside visual representation learning and demands significant extensions to graph representation learning, leading to our innovations below.

**Our Contributions.** In this paper, we have developed contrastive learning with augmentations for GNN pre-training to address the challenge of data heterogeneity in graphs. (i) Since data augmentations are the prerequisite for constrastive learning but are under-explored in graph-data [40], we first design four types of graph data augmentations, each of which imposes certain prior over graph data and parameterized for the extent and pattern. (ii) Utilizing them to obtain correlated views, we propose a novel graph contrastive learning framework (GraphCL) for GNN pre-training, so that representations invariant to specialized perturbations can be learned for diverse graph-structured data. Moreover, we show that GraphCL actually performs mutual information maximization, and the connection is drawn between GraphCL and recently proposed contrastive learning methods that we demonstrate that GraphCL can be rewritten as *a general framework* unifying a broad family of contrastive learning methods on graph-structured data. (iii) Systematic study is performed to assess the performance of contrasting different augmentations on various types of datasets, revealing the rationale of the performances and providing the guidance to adopt the framework for specific datasets. (iv) Experiments show that GraphCL achieves state-of-the-art performance in the settings of semi-supervised learning, unsupervised representation learning and transfer learning. It additionally boosts robustness against common adversarial attacks.

## 2 Related Work

**Graph neural networks.** In recent years, graph neural networks (GNNs) [1, 2, 3] have emerged as a promising approach for analyzing graph-structured data. They follow an iterative neighborhood aggregation (or message passing) scheme to capture the structural information within nodes' neighborhood. Let $\mathcal{G} = \{\mathcal{V}, \mathcal{E}\}$ denote an undirected graph, with $\boldsymbol{X} \in \mathbb{R}^{|\mathcal{V}| \times N}$ as the feature matrix where $\boldsymbol{x}_n = \boldsymbol{X}[n, :]^T$ is the $N$-dimensional attribute vector of the node $v_n \in \mathcal{V}$. Considering a $K$-layer GNN $f(\cdot)$, the propagation of the $k$th layer is represented as:

$$\boldsymbol{a}_n^{(k)} = \text{AGGREGATION}^{(k)}(\{\boldsymbol{h}_{n'}^{(k-1)} : n' \in \mathcal{N}(n)\}), \boldsymbol{h}_n^{(k)} = \text{COMBINE}^{(k)}(\boldsymbol{h}_n^{(k-1)}, \boldsymbol{a}_n^{(k)}), \quad (1)$$

where $\boldsymbol{h}_n^{(k)}$ is the embedding of the vertex $v_n$ at the $k$th layer with $\boldsymbol{h}_n^{(0)} = \boldsymbol{x}_n$, $\mathcal{N}(n)$ is a set of vertices adjacent to $v_n$, and $\text{AGGREGATION}^{(k)}(\cdot)$ and $\text{COMBINE}^{(k)}(\cdot)$ are component functions of the GNN layer. After the $K$-layer propagation, the output embedding for $\mathcal{G}$ is summarized on layer embeddings through the READOUT function. Then a multi-layer perceptron (MLP) is adopted for the graph-level downstream task (classification or regression):

$$f(\mathcal{G}) = \text{READOUT}(\{\boldsymbol{h}_n^{(k)} : v_n \in \mathcal{V}, k \in K\}), \boldsymbol{z}_{\mathcal{G}} = \text{MLP}(f(\mathcal{G})). \quad (2)$$

Various GNNs have been proposed [1, 2, 3], achieving state-of-the-art performance in graph tasks.

**Graph data augmentation.** Augmentation for graph-structured data still remains under-explored, with some work along these lines but requiring prohibitive additional computation cost [40]. Traditional self-training methods [40, 13] utilize the trained model to annotate unlabelled data; [41]

proposes to train a generator-classifier network in the adversarial learning setting to generate fake nodes; and [42, 43] generate adversarial perturbations to node feature over the graph structure.

**Pre-training GNNs.** Although (self-supervised) pre-training is a common and effective scheme for convolutional neural networks (CNNs) [16, 17, 18], it is rarely explored for GNNs. One exception [9] is restricted to studying pre-training strategies in the transfer learning setting, We argue that a pre-trained GNN is not easy to transfer, due to the diverse fields that graph-structured data source from. During transfer, substantial domain knowledge is required for both pre-training and downstream tasks, otherwise it might lead to negative transfer [9, 44].

**Contrastive learning.** The main idea of contrastive learning is to make representations agree with each other under proper transformations, raising a recent surge of interest in visual representation learning [45, 25, 26, 27, 18]. On a parallel note, for graph data, traditional methods trying to reconstruct the adjacency information of vertices [22, 23] can be treated as a kind of "local contrast", while over-emphasizing the proximity information at the expense of the structural information [24]. Motivated by [46, 47], [24, 21, 48] propose to perform contrastive learning between local and global representations to better capture structure information. However, graph contrastive learning has not been explored from the perspective of enforcing perturbation invariance as [27, 18] have done.

## 3 Methodology

### 3.1 Data Augmentation for Graphs

Data augmentation aims at creating novel and realistically rational data through applying certain transformation without affecting the semantics label. It still remains under-explored for graphs except some with expensive computation cost (see Sec. 2). We focus on graph-level augmentations. Given a graph $\mathcal{G} \in \{\mathcal{G}_m : m \in M\}$ in the dataset of $M$ graphs, we formulate the augmented graph $\hat{\mathcal{G}}$ satisfying: $\hat{\mathcal{G}} \sim q(\hat{\mathcal{G}}|\mathcal{G})$, where $q(\cdot|\mathcal{G})$ is the augmentation distribution conditioned on the original graph, which is pre-defined, representing the human prior for data distribution. For instance for image classification, the applications of rotation and cropping encode the prior that people will acquire the same classification-based semantic knowledge from the rotated image or its local patches [49, 50].

When it comes to graphs, the same spirit could be followed. However, one challenge as stated in Sec. 1 is that graph datasets are abstracted from diverse fields and therefore there may not be universally appropriate data augmentation as those for images. In other words, for different categories of graph datasets some data augmentations might be more desired than others. We mainly focus on three categories: biochemical molecules (e.g. chemical compounds, proteins) [9], social networks [1] and image super-pixel graphs [12]. Next, we propose four general data augmentations for graph-structured data and discuss the intuitive priors that they introduce.

**Table 1:** Overview of data augmentations for graphs.

| Data augmentation | Type | Underlying Prior |
|---|---|---|
| Node dropping | Nodes, edges | Vertex missing does not alter semantics. |
| Edge perturbation | Edges | Semantic robustness against connectivity variations. |
| Attribute masking | Nodes | Semantic robustness against losing partial attributes per node. |
| Subgraph | Nodes, edges | Local structure can hint the full semantics. |

**Node dropping.** Given the graph $\mathcal{G}$, node dropping will randomly discard certain portion of vertices along with their connections. The underlying prior enforced by it is that missing part of vertices does not affect the semantic meaning of $\mathcal{G}$. Each node's dropping probability follows a default i.i.d. uniform distribution (or any other distribution).

**Edge perturbation.** It will perturb the connectivities in $\mathcal{G}$ through randomly adding or dropping certain ratio of edges. It implies that the semantic meaning of $\mathcal{G}$ has certain robustness to the edge connectivity pattern variances. We also follow an i.i.d. uniform distribution to add/drop each edge.

**Attribute masking.** Attribute masking prompts models to recover masked vertex attributes using their context information, i.e., the remaining attributes. The underlying assumption is that missing partial vertex attributes does not affect the model predictions much.

**Subgraph.** This one samples a subgraph from $\mathcal{G}$ using random walk (the algorithm is summarized in Appendix A). It assumes that the semantics of $\mathcal{G}$ can be much preserved in its (partial) local structure.

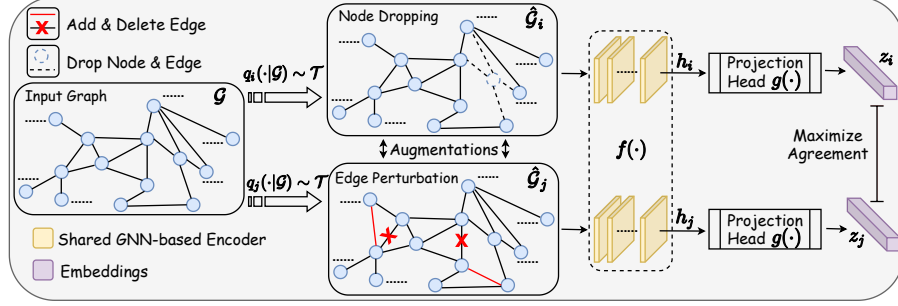

**Figure 1:** A framework of graph contrastive learning. Two graph augmentations $q_i(\cdot|\mathcal{G})$ and $q_j(\cdot|\mathcal{G})$ are sampled from an augmentation pool $\mathcal{T}$ and applied to input graph $\mathcal{G}$. A shared GNN-based encoder $f(\cdot)$ and a projection head $g(\cdot)$ are trained to maximize the agreement between representations $z_i$ and $z_j$ via a contrastive loss.

The default augmentation (dropping, perturbation, masking and subgraph) ratio is set at 0.2.

## 3.2 Graph Contrastive Learning

Motivated by recent contrastive learning developments in visual representation learning (see Sec. 2), we propose a graph contrastive learning framework (GraphCL) for (self-supervised) pre-training of GNNs. In graph contrastive learning, pre-training is performed through maximizing the agreement between two augmented views of the same graph via a contrastive loss in the latent space as shown in Fig. 1. The framework consists of the following four major components:

(1) **Graph data augmentation.** The given graph $\mathcal{G}$ undergoes graph data augmentations to obtain two correlated views $\hat{\mathcal{G}}_i, \hat{\mathcal{G}}_j$, as a positive pair, where $\hat{\mathcal{G}}_i \sim q_i(\cdot|\mathcal{G}), \hat{\mathcal{G}}_j \sim q_j(\cdot|\mathcal{G})$ respectively. For different domains of graph datasets, how to strategically select data augmentations matters (Sec. 4).

(2) **GNN-based encoder.** A GNN-based encoder $f(\cdot)$ (defined in (2)) extracts graph-level representation vectors $\boldsymbol{h}_i, \boldsymbol{h}_j$ for augmented graphs $\hat{\mathcal{G}}_i, \hat{\mathcal{G}}_j$. Graph contrastive learning does not apply any constraint on the GNN architecture.

(3) **Projection head.** A non-linear transformation $g(\cdot)$ named projection head maps augmented representations to another latent space where the contrastive loss is calculated, as advocated in [18]. In graph contrastive learning, a two-layer perceptron (MLP) is applied to obtain $\boldsymbol{z}_i, \boldsymbol{z}_j$.

(4) **Contrastive loss function.** A contrastive loss function $\mathcal{L}(\cdot)$ is defined to enforce maximizing the consistency between positive pairs $\boldsymbol{z}_i, \boldsymbol{z}_j$ compared with negative pairs. Here we utilize the normalized temperature-scaled cross entropy loss (NT-Xent) [51, 25, 52].

During GNN pre-training, a minibatch of $N$ graphs are randomly sampled and processed through contrastive learning, resulting in $2N$ augmented graphs and corresponding contrastive loss to optimize, where we re-annotate $z_i, z_j$ as $z_{n,i}, z_{n,j}$ for the $n$th graph in the minibatch. Negative pairs are not explicitly sampled but generated from the other $N - 1$ augmented graphs within the same minibatch as in [53, 18]. Denoting the cosine similarity function as $\mathrm{sim}(\boldsymbol{z}_{n,i}, \boldsymbol{z}_{n,j}) = \boldsymbol{z}_{n,i}^{\mathsf{T}} \boldsymbol{z}_{n,j} / \|\boldsymbol{z}_{n,i}\| \|\boldsymbol{z}_{n,j}\|$, NT-Xent for the $n$th graph is defined as:

$$\ell_n = -\log \frac{\exp(\mathrm{sim}(\boldsymbol{z}_{n,i}, \boldsymbol{z}_{n,j})/\tau)}{\sum_{n'=1,n'\neq n}^{N} \exp(\mathrm{sim}(\boldsymbol{z}_{n,i}, \boldsymbol{z}_{n',j})/\tau)}, \tag{3}$$

where $\tau$ denotes the temperature parameter. The final loss is computed across all positive pairs in the minibatch. The proposed graph contrastive learning is summarized in Appendix A.

**Discussion.** We first show that GraphCL can be viewed as one way of mutual information maximization between the latent representations of two kinds of augmented graphs. The full derivation is in Appendix F, with the loss form rewritten as below:

$$\ell = \mathbb{E}_{\mathbb{P}_{\hat{\mathcal{G}}_i}} \{ -\mathbb{E}_{\mathbb{P}_{(\hat{\mathcal{G}}_j|\hat{\mathcal{G}}_i)}} T(f_1(\hat{\mathcal{G}}_i), f_2(\hat{\mathcal{G}}_j)) + \log(\mathbb{E}_{\mathbb{P}_{\hat{\mathcal{G}}_j}} e^{T(f_1(\hat{\mathcal{G}}_i), f_2(\hat{\mathcal{G}}_j))}) \}. \tag{4}$$

The above loss essentially maximizes a lower bound of the mutual information between $\boldsymbol{h}_i = f_1(\hat{\mathcal{G}}_i), \boldsymbol{h}_j = f_2(\hat{\mathcal{G}}_j)$ that the compositions of $(f_1, \hat{\mathcal{G}}_i), (f_2, \hat{\mathcal{G}}_j)$ determine our desired views of

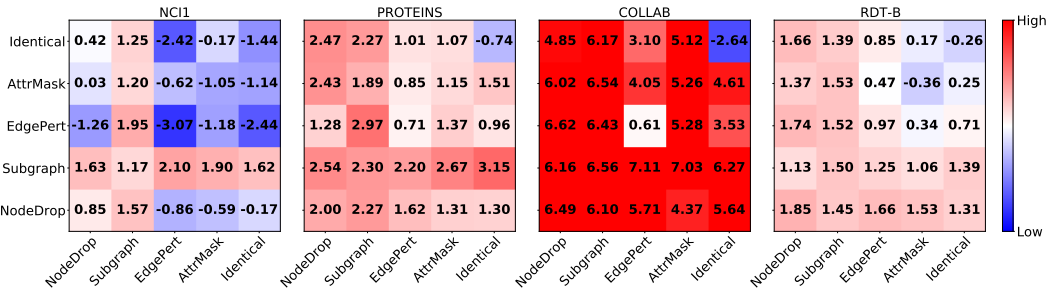

**Figure 2:** Semi-supervised learning accuracy gain (%) when contrasting different augmentation pairs, compared to training from scratch, under four datasets: NCI1, PROTEINS, COLLAB, and RDT-B. Pairing "Identical" stands for a no-augmentation baseline for contrastive learning, where the positive pair diminishes and the negative pair consists of two non-augmented graphs. Warmer colors indicate better performance gains. The baseline training-from-scratch accuracies are 60.72%, 70.40%, 57.46%, 86.63% for the four datasets respectively.

graphs. Furthermore, we draw the connection between GraphCL and recently proposed contrastive learning methods that we demonstrate that GraphCL can be rewrited as a general framework unifying a broad family of contrastive learning methods on graph-structured data, through reinterpreting (4). In our implementation, we choose $f_1 = f_2$ and generate $\hat{\mathcal{G}}_i, \hat{\mathcal{G}}_j$ through data augmentation, while with various choices of the compositions result in (4) instantiating as other specific contrastive learning algorithms including [54, 55, 56, 21, 57, 58, 59] also shown in in Appendix F.

## 4 The Role of Data Augmentation in Graph Contrastive Learning

In this section, we assess and rationalize the role of data augmentation for graph-structured data in our GraphCL framework. Various pairs of augmentation types are applied, as illustrated in Fig. 2, to three categories of graph datasets (Table 2, and we leave the discussion on

**Table 2:** Datasets statistics.

| Datasets | Category | Graph Num. | Avg. Node | Avg. Degree |
|---|---|---|---|---|
| NCI1 | Biochemical Molecules | 4110 | 29.87 | 1.08 |
| PROTEINS | Biochemical Molecules | 1113 | 39.06 | 1.86 |
| COLLAB | Social Networks | 5000 | 74.49 | 32.99 |
| RDT-B | Social Networks | 2000 | 429.63 | 1.15 |

superpixel graphs in Appendix C). Experiments are performed in the semi-supervised setting, following the pre-training & finetuning approach [18]. Detailed settings are in Appendix B.

### 4.1 Data Augmentations are Crucial. Composing Augmentations Benefits.

We first examine whether and when applying (different) data augmentations helps graph contrastive learning in general. We summarize the results in Fig. 2 using the accuracy gain compared to training from scratch (no pre-training). And we list the following **Obs**ervations.

**Obs. 1. Data augmentations are crucial in graph contrastive learning.** Without any data augmentation graph contrastive learning is not helpful and often worse compared with training from scratch, judging from the accuracy losses in the upper right corners of Fig. 2. In contrast, composing an original graph and its appropriate augmentation can benefit the downstream performance. Judging from the top rows or the right-most columns in Fig. 2, graph contrastive learning with single best augmentations achieved considerable improvement without exhaustive hyper-parameter tuning: 1.62% for NCI1, 3.15% for PROTEINS, 6.27% for COLLAB, and 1.66% for RDT-B.

The observation meets our intuition. Without augmentation, graphCL simply compares two original samples as a negative pair (with the positive pair loss becoming zero), leading to homogeneously pushes all graph representations away from each other, which is non-intuitive to justify. Importantly, when appropriate augmentations are applied, the corresponding priors on the data distribution are instilled, enforcing the model to learn representations invariant to the desired perturbations through maximizing the agreement between a graph and its augmentation.

**Obs. 2. Composing different augmentations benefits more.** Composing augmentation pairs of a graph rather than the graph and its augmentation further improves the performance: the maximum

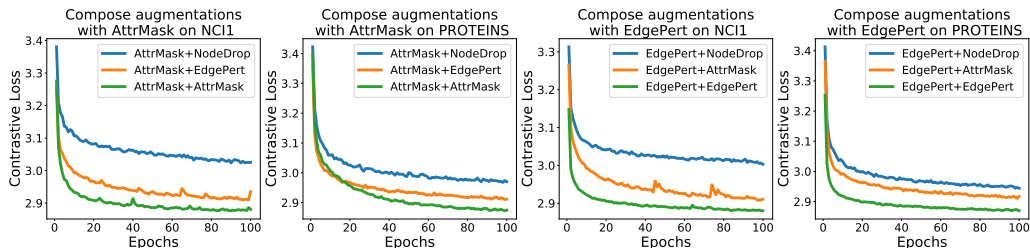

**Figure 3:** Contrastive loss curves for different augmentation pairs. In the two figures of the left attribute masking is contrasted with other augmentations and that of the right for edge perturbation, where contrasting the same augmentations always leads to the fastest loss descent.

accuracy gain was 2.10% for NCI1, 3.15% for PROTEINS, 7.11% for COLLAB, and 1.85% for RDT-B. Interestingly, applying augmentation pairs of the same type (see the diagonals of Fig. 2) does not usually lead to the best performance (except for node dropping), compared with augmentation pairs of different types (off-diagonals). Similar observations were made in visual representation learning [18]. As conjectured in [18], composing different augmentations avoids the learned features trivially overfitting low-level "shortcuts", making features more generalizable.

Here we make a similar conjecture that contrasting isogenous graph pairs augmented in different types presents a harder albeit more useful task for graph representation learning. We thus plot the contrastive loss curves composing various augmentations (except subgraph) together with attribute masking or edge perturbation for NCI1 and PROTEINS. Fig. 3 shows that, with augmentation pairs of different types, the contrastive loss always descents slower than it does with pairs of the same type, when the optimization procedure remains the same. This result indicates that composing augmentation pairs of different types does correspond to a "harder" contrastive prediction task. We will explore in Sec. 4.3 how to quantify a "harder" task in some cases and whether it always helps.

## 4.2 The Types, the Extent, and the Patterns of Effective Graph Augmentations

We then note that the (most) beneficial combinations of augmentation types can be dataset-specific, which matches our intuition as graph-structured data are of highly heterogeneous nature (see Sec. 1). We summarize our observations and derive insights below. And we further analyze the impact of the extent and/or the pattern of given types of graph augmentations.

**Obs. 3. Edge perturbation benefits social networks but hurts some biochemical molecules.** Edge perturbation as one of the paired augmentations improves the performances for social-network data COLLAB and ROT-B as well as biomolecule data PROTEINS, but hurts the other biomolecule data NCI1. We hypothesize that, compared to the case of social networks, the "semantemes" of some biomolecule data are more sensitive to individual edges. Specifically, a single-edge change in NCI1 corresponds to a removal or addition of a covalent bond, which can drastically change the identity and even the validity of a compound, let alone its property for the down-stream semantemes. In contrast the semantemes of social networks are more tolerant to individual edge perturbation [60, 61]. Therefore, for chemical compounds, edge perturbation demonstrates a prior that is conceptually incompatible with the domain knowledge and empirically unhelpful for down-stream performance.

We further examine whether the extent or strength of edge perturbation can affect the conclusion above. We evaluate the downstream performances on representative examples NCI1 and COLLAB. And we use the combination of the original graph ("identical") and edge perturbation of various ratios in our GraphCL framework. Fig. 4A shows that edge perturbation worsens the NCI1 performances regardless of augmentation strength, confirming that our earlier conclusion was insensitive to the extent of edge perturbation. Fig. 4B suggests that edge perturbation could improve the COLLAB performances more with increasing augmentation strength.

**Obs. 4. Applying attribute masking achieves better performance in denser graphs.** For the social network datasets, composing the identical graph and attribute masking achieves 5.12% improvement for COLLAB (with higher average degree) while only 0.17% for RDT-B. Similar observations are made for the denser PROTEINS versus NCI1. To assess the impact of augmentation strength on this observation, we perform similar experiments on RDT-B and COLLAB, by composing the

identical graph and its attributes masked to various extents. Fig. 4C and D show that, masking less for the very sparse RDT-B does not help, although masking more for the very dense COLLAB does.

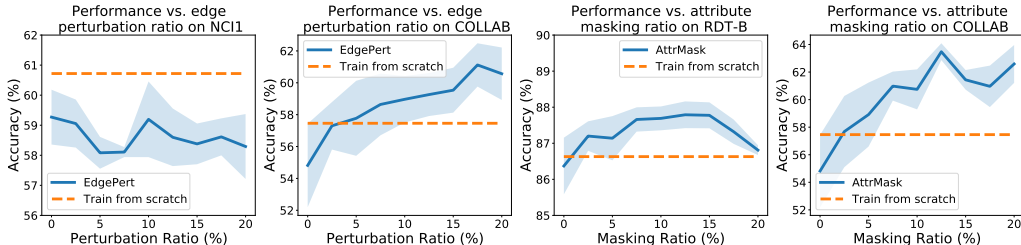

**Figure 4:** Performance versus augmentation strength. Left two figures implemented edge perturbation with different ratios. The right two figures apply attribute masking with different masking ratios.

We further hypothesize that masking patterns also matter, and masking more hub nodes with high degrees benefit denser graphs, because GNNs cannot reconstruct the missing information of isolated nodes, according to the message passing mechanism [62]. To test the hypothesis, we perform an experiment to mask nodes with more connections with higher probability on denser graphs PROTEINS and COLLAB. Specifically, we adopt a masking distribution $\deg_n^\alpha$ rather than the uniform distribution, where $\deg_n$ is the degree of vertex $v_n$ and $\alpha$ is the control factor. A positive $\alpha$ indicates more masking for high-degree nodes. Fig. 5C and D showing that, for very dense COLLAB, there is an apparent upward tendency on performance if masking nodes with more connections.

**Obs. 5. Node dropping and subgraph are generally beneficial across datasets.** Node dropping and subgraph, especially the latter, seem to be generally beneficial in our studied datasets. For node dropping, the prior that missing certain vertices (e.g. some hydrogen atoms in chemical compounds or edge users for social networks) does not alter the semantic information is emphasized, intuitively fitting for our cognition. For subgraph, previous works [20, 21] show that enforcing local (the subgraphs we extract) and global information consistency is helpful for representation learning, which explains the observation. Even for chemical compounds in NCI1, subgraphs can represent structural and functional "motifs" important for the down-stream semantemes.

We similarly examined the impact of node dropping patterns by adopting the non-uniform distribution as mentioned in changing attribute-masking patterns. Fig. 5B shows that, for the dense social-network COLLAB graphs, more GraphCL improvements were observed while dropping hub nodes more in the range considered. Fig. 5A shows that, for the not-so-dense PROTEINS graphs, changing the node-dropping distribution away from uniform does not necessarily help.

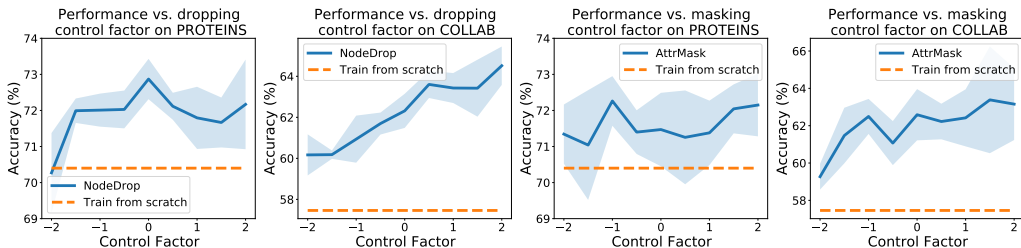

**Figure 5:** Performance versus augmentation patterns. Node dropping and attribute masking are performed with various control factors (negative to positive: dropping/masking more low-degree vertices to high-degree ones).

### 4.3 Unlike "Harder" Ones, Overly Simple Contrastive Tasks Do Not Help.

As discussed in Obs. 2, "harder" contrastive learning might benefit more, where the "harder" task is achieved by composing augmentations of different types. In this section we further explore quantifiable difficulty in relationship to parameterized augmentation strengths/patterns and assess the impact of the difficulty on performance improvement.

Intuitively, larger dropping/masking ratios or control factor $\alpha$ leads to harder contrastive tasks, which did result in better COLLAB performances (Fig. 4 and 5) in the range considered. Very small ratios or negative $\alpha$, corresponding to overly simple tasks, We also design subgraph variants of increasing difficulty levels and reach similar conclusions. More details are in Appendix D.

**Summary.** In total, we decide the augmentation pools for Section 5 as: node dropping and subgraph for biochemical molecules; all for dense social networks; and all except attribute masking for sparse social networks. Strengths or patterns are default even though varying them could help more.

## 5 Comparison with the State-of-the-art Methods

In this section, we compare our proposed (self-supervised) pre-training framework, GraphCL, with state-of-the-art methods (SOTAs) in the settings of semi-supervised, unsupervised [21] and transfer learning [9] on graph classification (for node classification experiments please refer to Appendix G). Dataset statistics and training details for the specific settings are in Appendix E.

**Semi-supervised learning.** We first evaluate our proposed framework in the semi-supervised learning setting on graph classification [63, 3] on the benchmark TUDataset [64]. Since pre-training & finetuning in semi-supervised learning for the graph-level task is unexplored before, we take two conventional network embedding methods as pre-training tasks for comparison: adjacency information reconstruction (we refer to GAE [22] for implementation) and local & global representation consistency enforcement (refer to Infomax [21] for implementation). Besides, the performance of training from scratch and that with augmentation (without contrasting) is also reported. We adopt graph convolutional network (GCN) with the default setting in [63] as the GNN-based encoder which achieves comparable SOTA performance in the fully-supervised setting. Table 3 shows that GraphCL outperforms traditional pre-training schemes.

**Table 3:** Semi-supervised learning with pre-training & finetuning. Red numbers indicate the best performance and the number that overlap with the standard deviation of the best performance (comparable ones). 1% or 10% is label rate; baseline and Aug. represents training from scratch without and with augmentations, respectively.

| Dataset | NCI1 | PROTEINS | DD | COLLAB | RDT-B | RDT-M5K | GITHUB | MNIST | CIFAR10 |
|---|---|---|---|---|---|---|---|---|---|
| 1% baseline | 60.72±0.45 | - | - | 57.46±0.25 | - | - | 54.25±0.22 | 60.39±1.95 | 27.36±0.75 |
| 1% Aug. | 60.49±0.46 | - | - | 58.40±0.97 | - | - | 56.36±0.42 | 67.43±0.36 | 27.39±0.44 |
| 1% GAE | 61.63±0.84 | - | - | 63.20±0.67 | - | - | 59.44±0.44 | 57.58±2.07 | 21.09±0.53 |
| 1% Infomax | 62.72±0.65 | - | - | 61.70±0.77 | - | - | 58.99±0.50 | 63.24±0.78 | 27.86±0.43 |
| 1% GraphCL | 62.55±0.86 | - | - | 64.57±1.15 | - | - | 58.56±0.59 | 83.41±0.33 | 30.01±0.84 |
| 10% baseline | 73.72±0.24 | 70.40±1.54 | 73.56±0.41 | 73.71±0.27 | 86.63±0.27 | 51.33±0.44 | 60.87±0.17 | 79.71±0.65 | 35.78±0.81 |
| 10% Aug. | 73.59±0.32 | 70.29±0.64 | 74.30±0.81 | 74.19±0.13 | 87.74±0.39 | 52.01±0.20 | 60.91±0.32 | 83.99±2.19 | 34.24±2.62 |
| 10% GAE | 74.36±0.24 | 70.51±0.17 | 74.54±0.68 | 75.09±0.19 | 87.69±0.40 | 53.58±0.13 | 63.89±0.52 | 86.67±0.93 | 36.35±1.04 |
| 10% Infomax | 74.86±0.26 | 72.27±0.40 | 75.78±0.34 | 73.76±0.29 | 88.66±0.95 | 53.61±0.31 | 65.21±0.88 | 83.34±0.24 | 41.07±0.48 |
| 10% GraphCL | 74.63±0.25 | 74.17±0.34 | 76.17±1.37 | 74.23±0.21 | 89.11±0.19 | 52.55±0.45 | 65.81±0.79 | 93.11±0.17 | 43.87±0.77 |

**Unsupervised representation learning.** Furthermore, GraphCL is evaluated in the unsupervised representation learning following [65, 21], where unsupervised methods generate graph embeddings that are fed into a down-stream SVM classifier [21]. Aside from SOTA graph kernel methods that graphlet kernel (GL), Weisfeiler-Lehman sub-tree kernel (WL) and deep graph kernel (DGK), we also compare with four unsupervised graph-level representation learning methods as node2vec [66], sub2vec [67], graph2vec [65] and InfoGraph [21]. We adopt graph isomorphism network (GIN) with the default setting in [21] as the GNN-based encoder which is SOTA in representation learning. Table 4 shows GraphCL outperforms in most cases except on datasets with small graph size (e.g. MUTAG and IMDB-B consists of graphs with average node number less than 20).

**Table 4:** Comparing classification accuracy on top of graph representations learned from graph kernels, SOTA representation learning methods, and GIN pre-trained with GraphCL. The compared numbers are from the corresponding papers under the same experiment setting.

| Dataset | NCI1 | PROTEINS | DD | MUTAG | COLLAB | RDT-B | RDT-M5K | IMDB-B |
|---|---|---|---|---|---|---|---|---|
| GL | - | - | - | 81.66±2.11 | - | 77.34±0.18 | 41.01±0.17 | 65.87±0.98 |
| WL | 80.01±0.50 | 72.92±0.56 | - | 80.72±3.00 | - | 68.82±0.41 | 46.06±0.21 | 72.30±3.44 |
| DGK | 80.31±0.46 | 73.30±0.82 | - | 87.44±2.72 | - | 78.04±0.39 | 41.27±0.18 | 66.96±0.56 |
| node2vec | 54.89±1.61 | 57.49±3.57 | - | 72.63±10.20 | - | - | - | - |
| sub2vec | 52.84±1.47 | 53.03±5.55 | - | 61.05±15.80 | - | 71.48±0.41 | 36.68±0.42 | 55.26±1.54 |
| graph2vec | 73.22±1.81 | 73.30±2.05 | - | 83.15±9.25 | - | 75.78±1.03 | 47.86±0.26 | 71.10±0.54 |
| InfoGraph | 76.20±1.06 | 74.44±0.31 | 72.85±1.78 | 89.01±1.13 | 70.65±1.13 | 82.50±1.42 | 53.46±1.03 | 73.03±0.87 |
| GraphCL | 77.87±0.41 | 74.39±0.45 | 78.62±0.40 | 86.80±1.34 | 71.36±1.15 | 89.53±0.84 | 55.99±0.28 | 71.14±0.44 |

**Transfer learning.** Lastly, experiments are performed on transfer learning on molecular property prediction in chemistry and protein function prediction in biology following [9], which pre-trains and finetunes the model in different datasets to evaluate the transferability of the pre-training scheme. We adopt GIN with the default setting in [9] as the GNN-based encoder which is SOTA in transfer

learning. Experiments are performed for 10 times with mean and standard deviation of ROC-AUC scores (%) reported as [9]. Although there is no universally beneficial pre-training scheme especially for the out-of-distribution scenario in transfer learning (Sec. 1), Table 5 shows that GraphCL still achieves SOTA performance on 5 of 9 datasets compared to the previous best schemes.

**Table 5:** Transfer learning comparison with different manually designed pre-training schemes, where the compared numbers are from [9].

| Dataset | BBBP | Tox21 | ToxCast | SIDER | ClinTox | MUV | HIV | BACE | PPI |
|---|---|---|---|---|---|---|---|---|---|
| No Pre-Train | 65.8±4.5 | 74.0±0.8 | 63.4±0.6 | 57.3±1.6 | 58.0±4.4 | 71.8±2.5 | 75.3±1.9 | 70.1±5.4 | 64.8±1.0 |
| Infomax | 68.8±0.8 | 75.3±0.5 | 62.7±0.4 | 58.4±0.8 | 69.9±3.0 | 75.3±2.5 | 76.0±0.7 | 75.9±1.6 | 64.1±1.5 |
| EdgePred | 67.3±2.4 | 76.0±0.6 | 64.1±0.6 | 60.4±0.7 | 64.1±3.7 | 74.1±2.1 | 76.3±1.0 | 79.9±0.9 | 65.7±1.3 |
| AttrMasking | 64.3±2.8 | 76.7±0.4 | 64.2±0.5 | 61.0±0.7 | 71.8±4.1 | 74.7±1.4 | 77.2±1.1 | 79.3±1.6 | 65.2±1.6 |
| ContextPred | 68.0±2.0 | 75.7±0.7 | 63.9±0.6 | 60.9±0.6 | 65.9±3.8 | 75.8±1.7 | 77.3±1.0 | 79.6±1.2 | 64.4±1.3 |
| GraphCL | 69.68±0.67 | 73.87±0.66 | 62.40±0.57 | 60.53±0.88 | 75.99±2.65 | 69.80±2.66 | 78.47±1.22 | 75.38±1.44 | 67.88±0.85 |

**Adversarial robustness.** In addition to generalizability, we claim that GNNs also gain robustness using GraphCL. The experiments are performed on synthetic data to classify the component number in graphs, facing the RandSampling, GradArgmax and RL-S2V attacks following the default setting in [60]. Structure2vec [68] is adopted as the GNN-based encoder as in [60]. Table 6 shows that GraphCL boosts GNN robustness compared with training from scratch, under three evasion attacks.

**Table 6:** Adversarial performance under three adversarial attacks for GNN with different depth (following the protocol in [60]). Red numbers indicate the best performance.

| Methods | Two-Layer | | Three-Layer | | Four-Layer | |
|---|---|---|---|---|---|---|
| | No Pre-Train | GraphCL | No Pre-Train | GraphCL | No Pre-Train | GraphCL |
| Unattack | 93.20 | 94.73 | 98.20 | 98.33 | 98.87 | 99.00 |
| RandSampling | 78.73 | 80.68 | 92.27 | 92.60 | 95.13 | 97.40 |
| GradArgmax | 69.47 | 69.26 | 64.60 | 89.33 | 95.80 | 97.00 |
| RL-S2V | 42.93 | 42.20 | 41.93 | 61.66 | 70.20 | 84.86 |

## 6  Conclusion

In this paper, we perform explicit study to explore contrastive learning for GNN pre-training, facing the unique challenges in graph-structured data. Firstly, several graph data augmentations are proposed with the discussion of each of which on introducing certain human prior of data distribution. Along with new augmentations, we propose a novel graph contrastive learning framework (GraphCL) for GNN pre-training to facilitate invariant representation learning along with rigorous theoretical analysis. We systematically assess and analyze the influence of data augmentations in our proposed framework, revealing the rationale and guiding the choice of augmentations. Experiment results verify the state-of-the-art performance of our proposed framework in both generalizability and robustness.

## Broader Impact

Empowering deep learning for reasoning and predicting over graph-structured data is of broad interests and wide applications, such as recommendation systems, neural architecture search, and drug discovery. The proposed graph contrastive learning framework with augmentations contributes a general framework that can potentially benefit the effectiveness and efficiency of graph neural networks through model pre-training. The numerical results and analyses would also inspire the design of proper augmentations toward positive knowledge transfer on downstream tasks.

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
