[Supplementary Material]

# Graph Contrastive Learning with Augmentations
## (Appendix)

**Yuning You[1*], Tianlong Chen[2*], Yongduo Sui[3], Ting Chen[4], Zhangyang Wang[2], Yang Shen[1]**
[1]Texas A&M University, [2]University of Texas at Austin,
[3]University of Science and Technology of China, [4]Google Research, Brain Team
{yuning.you,yshen}@tamu.edu, {tianlong.chen,atlaswang}@utexas.edu
syd2019@mail.ustc.edu.cn, iamtingchen@google.com

## A   Contrastive Learning and Augmentation Algorithms

---
**Algorithm 1** Graph Contrastive Learning

---
**Initialize:** Data $\{\mathcal{G}_m : m \in M\}$, $f(\cdot)$, $g(\cdot)$, $\mathcal{T}$
 1: **for** sampled minibatch of data $\{\mathcal{G}_n : n \in N\}$ **do**
 2:     **for** $n = 1$ **to** $N$ **do**
 3:         Sample $q_i, q_j$ from $\mathcal{T}$
 4:         $\hat{\mathcal{G}}_{n,i} \sim q_i(\cdot|\mathcal{G}_n)$ # $1^{\text{st}}$ augmentation
 5:         $\boldsymbol{h}_{n,i} = f(\hat{\mathcal{G}}_{n,i})$
 6:         $\boldsymbol{z}_{n,i} = g(\boldsymbol{h}_{n,i})$
 7:         $\hat{\mathcal{G}}_{n,j} \sim q_j(\cdot|\mathcal{G}_n)$ # $2^{\text{nd}}$ augmentation
 8:         $\boldsymbol{h}_{n,j} = f(\hat{\mathcal{G}}_{n,j})$
 9:         $\boldsymbol{z}_{n,j} = g(\boldsymbol{h}_{n,j})$
10:     **end for**
11:     **define** $\ell_n = -\log \frac{\exp(\text{sim}(\boldsymbol{z}_{n,i}, \boldsymbol{z}_{n,j})/\tau)}{\sum_{n'=1, n' \neq n}^{N} \exp(\text{sim}(\boldsymbol{z}_{n,i}, \boldsymbol{z}_{n',j})/\tau)}$
12:     $\ell = \frac{1}{N} \sum_{n=1}^{N} \ell_n$
13:     # Asymmetric and simplified compared to the SimCLR loss
14:     Update encoder $f(\cdot)$ and $g(\cdot)$ to minimize $\ell$ # maximize agreement
15: **end for**
16: **return** Encoder $f(\cdot)$

---

---
**Algorithm 2** Data Augmentation: Subgraph

---
**Initialize:** Graph $\mathcal{G} = \{\mathcal{V}, \mathcal{E}\}$, augmentation ratio $k$, sampled graph $\mathcal{G}_{\text{samp}} = \{\mathcal{V}_{\text{samp}}, \mathcal{E}_{\text{samp}}\}$ where
    $\mathcal{V}_{\text{samp}} = \mathcal{E}_{\text{samp}} = \emptyset$, neighbor vertex set $\mathcal{V}_{\text{neigh}} = \emptyset$
 1: Sample a node $v \in \mathcal{V}$, such that $\mathcal{V}_{\text{samp}} = \{v\}$ and $\mathcal{V}_{\text{neigh}} = \mathcal{N}(v)$
 2: **while** $|\mathcal{V}_{\text{samp}}| \leq k|\mathcal{V}|$ **do**
 3:     Sample a node $v \in \mathcal{V}_{\text{neigh}}$
 4:     **if** $v \in \mathcal{V}_{\text{samp}}$ **then**
 5:         Continue
 6:     **end if**
 7:     Update $\mathcal{V}_{\text{samp}} = \mathcal{V}_{\text{samp}} \cup \{v\}$, $\mathcal{V}_{\text{neigh}} = \mathcal{V}_{\text{neigh}} \cup \mathcal{N}(v)$
 8: **end while**
 9: Update $\mathcal{E}_{\text{samp}} = \{e | e \in \mathcal{E} \text{ and } (e[0] \in \mathcal{V}_{\text{samp}} \text{ or } e[1] \in \mathcal{V}_{\text{samp}})\}$
10: **return** $\mathcal{G}_{\text{samp}}$

---

---
**Algorithm 3** Data Augmentation: Subgraph-W
---
**Initialize:** Graph $\mathcal{G} = \{\mathcal{V}, \mathcal{E}\}$, augmentation ratio $k$, sampled graph $\mathcal{G}_{\text{samp}} = \{\mathcal{V}_{\text{samp}}, \mathcal{E}_{\text{samp}}\}$ where
$\quad \mathcal{V}_{\text{samp}} = \mathcal{E}_{\text{samp}} = \emptyset$, neighbor vertex set $\mathcal{V}_{\text{neigh}} = \emptyset$
1: Sample a node $v \in \mathcal{V}$, such that $\mathcal{V}_{\text{samp}} = \{v\}$ and $\mathcal{V}_{\text{neigh}} = \mathcal{N}(v)$
2: **while** $|\mathcal{V}_{\text{samp}}| \leq k|\mathcal{V}|$ **do**
3: $\quad$ Update $\mathcal{V}_{\text{samp}} = \mathcal{V}_{\text{samp}} \cup \mathcal{V}_{\text{neigh}}$, $\mathcal{V}_{\text{neigh}} = \cup_{v \in \mathcal{V}_{\text{samp}}} \mathcal{N}(v)$
4: **end while**
5: Update $\mathcal{E}_{\text{samp}} = \{e | e \in \mathcal{E} \text{ and } (e[0] \in \mathcal{V}_{\text{samp}} \text{ or } e[1] \in \mathcal{V}_{\text{samp}})\}$
6: **return** $\mathcal{G}_{\text{samp}}$
---

---
**Algorithm 4** Data Augmentation: Subgraph-D
---
**Initialize:** Graph $\mathcal{G} = \{\mathcal{V}, \mathcal{E}\}$, augmentation ratio $k$, sampled graph $\mathcal{G}_{\text{samp}} = \{\mathcal{V}_{\text{samp}}, \mathcal{E}_{\text{samp}}\}$ where
$\quad \mathcal{V}_{\text{samp}} = \mathcal{E}_{\text{samp}} = \emptyset$, neighbor vertex set $\mathcal{V}_{\text{neigh}} = \emptyset$
1: Sample a node $v \in \mathcal{V}$, such that $\mathcal{V}_{\text{samp}} = \{v\}$ and $\mathcal{V}_{\text{neigh}} = \mathcal{N}(v)$
2: **while** $|\mathcal{V}_{\text{samp}}| \leq k|\mathcal{V}|$ **do**
3: $\quad$ Sample a node $v \in \mathcal{V}_{\text{neigh}}$
4: $\quad$ **if** $v \in \mathcal{V}_{\text{samp}}$ **then**
5: $\quad\quad$ Continue
6: $\quad$ **end if**
7: $\quad$ Update $\mathcal{V}_{\text{samp}} = \mathcal{V}_{\text{samp}} \cup \{v\}$, $\mathcal{V}_{\text{neigh}} = \mathcal{N}(v)$
8: **end while**
9: Update $\mathcal{E}_{\text{samp}} = \{e | e \in \mathcal{E} \text{ and } (e[0] \in \mathcal{V}_{\text{samp}} \text{ or } e[1] \in \mathcal{V}_{\text{samp}})\}$
10: **return** $\mathcal{G}_{\text{samp}}$
---

## B  Detailed Settings for Augmentation Experiments (Section 4 in Main Text)

We evaluate our proposed framework with different augmentation pairs in the semi-supervised learning setting on graph classification [1] via pre-training & finetuing where pre-training is performed with 100 epochs, 0.001 learning rate, and finetuning follows the 10-fold evaluation finetuning in [2] that achieves the comparable SOTA performance in the fully-supervised setting. Graph convolutional network (GCN) is adopted as the GNN-based encoder also following [2]. Experiments are performed with 1% (if there are over 10 samples for each class) and 10% label rate for 5 times with mean and standard deviation of accuracies (%) reported.

## C  Graph Contrastive Learning for Superpixel Graphs

**Table S1:** Superpixel graph dataset statistics.

| Datasets | Category | Graph Num. | Avg. Node | Avg. Degree |
|---|---|---|---|---|
| MNIST | Superpixel Graphs | 70000 | 70.57 | 8 |

Superpixel graphs (statistics in Table S1) gain from all augmentations except attribute masking as shown in Figure S1. For node dropping, it corresponds to pixel discarding and for subgraph to cropping, which are already shown as useful augmentations in images [3]. Surprisingly, attribute masking corresponding to image completion hurts the performance, which might result from our implementation: node attributes of superpixel graphs contain information of pixel value and location, and we might only mask the pixel value part rather than all analog to image completion. We do not find a related augmentation with edge perturbation and leave it for future work.

## D  Difficulty of Contrastive Tasks v.s. Semi-Supervised Performance

We first note that, for edge perturbation, attribute masking, and node dropping, their extents could be an indicator of the difficulty for corresponding contrastive tasks. We observed earlier in Figure 4

**Figure S1:** Semi-supervised learning accuracy gain (%) when contrasting different augmentation pairs, compared to training from scratch under MNIST. Pairing "Identical" stands for a no-augmentation baseline for contrastive learning, where the positive pair diminishes and the negative pair consists of two non-augmented graphs. Warmer colors indicate better performance gains. The baseline training-from-scratch accuracy is 79.71%.

(main text) that, at least for COLLAB, properly increasing the extents could enhance the downstream performances.

We also note that, for attribute masking and node dropping, their patterns could correlate the difficulty as well. With larger control factor $\alpha$ in masking/dropping distribution, the vertices with more connections are masked/dropped with higher probability, intuitively leading to a "harder" task. We again observed in Figure 5 (main text) that at least COLLAB performances benefited from the harder task with the pattern change, while overly simple contrastive tasks with very negative $\alpha$ would not help.

**Table S2:** Performance on contrastive learning with different implemented subgraph. The intuitively simplest subgraph-W performs the worst among the three.

| Augmentations | Subgraph-W | Subgraph | Subgraph-D |
|---|---|---|---|
| PROTEINS | 71.50±0.85 | 72.67±0.60 | 72.74±0.56 |
| COLLAB | 57.66±1.64 | 63.63±1.20 | 65.47±1.43 |

For subgraph, we propose the following variants with difficulty levels. Contrastive learning with subgraphs sampled via depth-first-search (DFS) encouraged random walk is more difficult than that via width-first-search (WFS) encouraged, since the latter preserves more structure information (connections) to assist GNNs to recover semantic information [4]. Notice that our default subgraph encourages neither DFS nor WFS, and therefore we additionally proposed subgraph-D(FS) and subgraph-W(FS) (Algorithms summarized in Appendix A) with the intuitive difficulty rank: subgraph-W < subgraph < subgraph-D. Experiments on PROTEINS and COLLAB in Table S2 agrees with our previous conjecture, that the simplest subgraph-W yields the worst performance among the three.

# E   Datasets and Training in Various Settings (Section 5 in Main Text))

**Semi-supervised Learning**

For all datasets we perform experiments with 1% (if there are over 10 samples for each class) and 10% label rate for 5 times, each of which corresponds to a 10-fold evaluation as [2], with mean and standard deviation of accuracies (%) reported. For pre-training, learning rate is tuned in {0.01, 0.001, 0.0001} and epoch number in {20, 40, 60, 80, 100} where grid search is performed. We follow

**Table S3:** Datasets statistics for semi-supervised learning and unsupervised representation learning.

| Datasets | Category | Graph Num. | Avg. Node | Avg. Degree |
|----------|----------|------------|-----------|-------------|
| NCI1 | Biochemical Molecules | 4110 | 29.87 | 1.08 |
| PROTEINS | Biochemical Molecules | 1113 | 39.06 | 1.86 |
| DD | Biochemical Molecules | 1178 | 284.32 | 715.66 |
| MUTAG | Biochemical Molecules | 188 | 17.93 | 19.79 |
| COLLAB | Social Networks | 5000 | 74.49 | 32.99 |
| RDT-B | Social Networks | 2000 | 429.63 | 1.15 |
| RDB-M | Social Networks | 2000 | 429.63 | 497.75 |
| GITHUB | Social Networks | 4999 | 508.52 | 594.87 |
| IMDB-B | Social Networks | 1000 | 19.77 | 96.53 |
| MNIST | Superpixel Graphs | 70000 | 70.57 | 8 |
| CIFAR10 | Superpixel Graphs | 60000 | 117.63 | 8 |

the default setting in [2] for finetuning that achieves the SOTA performance in the fully-supervised setting.

**Unsupervised Representation Learning**

Experiments are performed for 5 times each of which corresponds to a 10-fold evaluation as [5], with mean and standard deviation of accuracies (%) reported.

**Transfer Learning**

**Table S4:** Datasets statistics for transfer learning.

| Datasets | Category | Utilization | Graph Num. | Avg. Node | Avg. Degree |
|----------|----------|-------------|------------|-----------|-------------|
| ZINC-2M | Biochemical Molecules | Pre-Training | 2000000 | 26.62 | 57.72 |
| PPI-306K | Protein-Protein Intersection Networks | Pre-Training | 306925 | 39.82 | 729.62 |
| BBBP | Biochemical Molecules | Finetuning | 2039 | 24.06 | 51.90 |
| Tox21 | Biochemical Molecules | Finetuning | 7831 | 18.57 | 38.58 |
| ToxCast | Biochemical Molecules | Finetuning | 8576 | 18.78 | 38.52 |
| SIDER | Biochemical Molecules | Finetuning | 1427 | 33.64 | 70.71 |
| ClinTox | Biochemical Molecules | Finetuning | 1477 | 26.15 | 55.76 |
| MUV | Biochemical Molecules | Finetuning | 93087 | 24.23 | 52.55 |
| HIV | Biochemical Molecules | Finetuning | 41127 | 25.51 | 54.93 |
| BACE | Biochemical Molecules | Finetuning | 1513 | 34.08 | 73.71 |
| PPI | Protein-Protein Intersection Networks | Finetuning | 88000 | 49.35 | 890.77 |

# F   Theoretical Justification

**Mutual information maximization.** We first conceptually depict the essence of our framework, rigorously showing that GraphCL can be viewed as a kind of mutual information maximization between the latent representations of two kinds of augmented graphs. We rewrite GraphCL loss for each data point as:

$$\ell_n = -\log \frac{\exp(\mathrm{sim}(\boldsymbol{z}_{n,i}, \boldsymbol{z}_{n,j})/\tau)}{\sum_{n'=1, n'\neq n}^{N} \exp(\mathrm{sim}(\boldsymbol{z}_{n,i}, \boldsymbol{z}_{n',j})/\tau)}, \tag{1}$$

which can be rewrited for a batch of graphs as:

$$\ell = -\frac{1}{N}\sum_{n=1}^{N}\log\frac{\exp(\mathrm{sim}(\boldsymbol{z}_{n,i},\boldsymbol{z}_{n,j})/\tau)}{\sum_{n'=1,n'\neq n}^{N}\exp(\mathrm{sim}(\boldsymbol{z}_{n,i},\boldsymbol{z}_{n',j})/\tau)}$$

$$= -\frac{1}{N}\sum_{n=1}^{N}[\mathrm{sim}(\boldsymbol{z}_{n,i},\boldsymbol{z}_{n,j})/\tau - \log(\sum_{n'=1,n'\neq n}^{N}\exp(\mathrm{sim}(\boldsymbol{z}_{n,i},\boldsymbol{z}_{n',j})/\tau))]$$

$$= -\frac{1}{N}\sum_{n=1}^{N}\frac{\mathrm{sim}(g(f(\hat{\mathcal{G}}_{n,i})),g(f(\hat{\mathcal{G}}_{n,j})))}{\tau} + \frac{1}{N}\sum_{n=1}^{N}\log(\sum_{n'=1,n'\neq n}^{N}\exp(\frac{\mathrm{sim}(g(f(\hat{\mathcal{G}}_{n,i})),g(f(\hat{\mathcal{G}}_{n,j})))}{\tau})),$$
(2)

We rewrite (2) as the expectation form (and therefore remove the subscript $n$):

$$\ell = -\mathbb{E}_{\mathbb{P}_{(\hat{\mathcal{G}}_i,\hat{\mathcal{G}}_j)}}\frac{\mathrm{sim}(g(f(\hat{\mathcal{G}}_i)),g(f(\hat{\mathcal{G}}_j)))}{\tau} + \mathbb{E}_{\mathbb{P}_{\hat{\mathcal{G}}_i}}\log(\mathbb{E}_{\mathbb{P}_{\hat{\mathcal{G}}_j}}\exp(\frac{\mathrm{sim}(g(f(\hat{\mathcal{G}}_i)),g(f(\hat{\mathcal{G}}_j)))}{\tau})) - \log N$$

$$= \mathbb{E}_{\mathbb{P}_{\hat{\mathcal{G}}_i}}\{-\mathbb{E}_{\mathbb{P}_{(\hat{\mathcal{G}}_j|\hat{\mathcal{G}}_i)}}\frac{\mathrm{sim}(g(f(\hat{\mathcal{G}}_i)),g(f(\hat{\mathcal{G}}_j)))}{\tau} + \log(\mathbb{E}_{\mathbb{P}_{\hat{\mathcal{G}}_j}}\exp(\frac{\mathrm{sim}(g(f(\hat{\mathcal{G}}_i)),g(f(\hat{\mathcal{G}}_j)))}{\tau}))\} - \log N$$

$$= \mathbb{E}_{\mathbb{P}_{\hat{\mathcal{G}}_i}}\{-\mathbb{E}_{\mathbb{P}_{(\hat{\mathcal{G}}_j|\hat{\mathcal{G}}_i)}}T(f(\hat{\mathcal{G}}_i),f(\hat{\mathcal{G}}_j)) + \log(\mathbb{E}_{\mathbb{P}_{\hat{\mathcal{G}}_j}}e^{T(f(\hat{\mathcal{G}}_i),f(\hat{\mathcal{G}}_j))})\} - \log N,$$
(3)

where $\mathbb{P}_{(\hat{\mathbb{G}}_i,\hat{\mathbb{G}}_j)}, \mathbb{P}_{(\hat{\mathbb{G}}_j|\hat{\mathbb{G}}_i)}, \mathbb{P}_{\hat{\mathbb{G}}_i}$ are respectively the joint, conditional and marginal distribution of augmented graphs, and $T(\cdot,\cdot)$ is a learnable score function that we parametrize with the similarity function $\mathrm{sim}(\cdot,\cdot)$, temperature factor $\tau$ and the projection head $g(\cdot)$. Thus, (3) fits the formulation of the InfoNCE loss [6, 7] such that minimizing (3) is equivalent to *maximizing a lower bound of the mutual information* between the latent representations of two views of graphs $\boldsymbol{h}_i = f(\hat{\mathcal{G}}_i), \boldsymbol{h}_j = f(\hat{\mathcal{G}}_j)$. We would like to highlight the crucial role of the projection head in the framework, that provides the learnable weights to construct a function space for the score function, to reach a much tighter lower bound compared with dropping the projection head (the key role of it is also empirically verified in [3]).

**General framework.** Furthermore, we draw the connection between GraphCL and recently proposed contrastive learning methods that we demonstrate that GraphCL can be rewrited as a general framework unifying a broad family of contrastive learning methods on graph-structured data, through our rewriting (3) as (we neglect $\log N$ for simplicity):

$$\ell = \mathbb{E}_{\mathbb{P}_{\hat{\mathcal{G}}_i}}\{-\mathbb{E}_{\mathbb{P}_{(\hat{\mathcal{G}}_j|\hat{\mathcal{G}}_i)}}T(f_1(\hat{\mathcal{G}}_i),f_2(\hat{\mathcal{G}}_j)) + \log(\mathbb{E}_{\mathbb{P}_{\hat{\mathcal{G}}_j}}e^{T(f_1(\hat{\mathcal{G}}_i),f_2(\hat{\mathcal{G}}_j))})\},$$
(4)

where we maximize a lower bound of the mutual information between $\boldsymbol{h}_i = f_1(\hat{\mathcal{G}}_i), \boldsymbol{h}_j = f_2(\hat{\mathcal{G}}_j)$ that the compositions of $(f_1,\hat{\mathcal{G}}_i),(f_2,\hat{\mathcal{G}}_j)$ determine our desired views of graphs. In our implementation, we choose $f_1 = f_2$ and generate $\hat{\mathcal{G}}_i,\hat{\mathcal{G}}_j$ through data augmentation, while with various choices of the compositions result in (4) instantiating as other specific contrastive learning algorithms including [8, 9, 10, 5, 11, 12, 13].

- **DGI, HDGI, DMGI** [8, 9, 10]. DGI intends to maximize the agreement between the local and global representations. Thus, it sets $f_1$ a GNN encoder, $f_2$ is the concatenation of $f_1$ and a node pooling layer, and $\hat{\mathcal{G}}_i = \hat{\mathcal{G}}_j = \mathcal{G}$. HDGI is an application of DGI in heterogenous graphs that the GNN encoder $f_1$ is a heterogenous GNN. DMGI is an extension of DGI into multiplex networks that perform DGI with multiple relationship types and the final representations are aggregated with multiple learned features.

- **InfoGraph** [5]. InfoGraph is an extension of DGI in graph-level representation learning, which is aimed at optimize the similarity between node embeddings and graph embeddings, and there for it sets $f_1 = f_2$ as a GNN encoder, $\hat{\mathcal{G}}_i = \mathcal{G}$ is the original graph and $\hat{\mathcal{G}}_j$ is the sampled subgraph.

- **GMI** [11]. GMI intends to maximize the agreement between the raw node & edge features and the encoded node & edge features, and therefore in the node loss $\ell_{\text{node}}$ it sets $\hat{\mathcal{G}}_{i,\text{node}} = \hat{\mathcal{G}}_{j,\text{node}} = \mathcal{G}$, $f_{1,\text{node}}(\cdot) = I(\cdot)$ is the identical function and $f_{2,\text{node}}(\cdot)$ is a GNN encoder, and similar for the edge loss $\ell_{\text{edge}}$ where $f_{2,\text{node}}(\cdot), f_{2,\text{edge}}(\cdot)$ share weights. The final loss is expressed as $\ell = \ell_{\text{node}} + \ell_{\text{edge}}$.

# G Experiments on Node Classification

**GraphCL in node classification.** GraphCL is also evaluated in the unsupervised representation learning in node classification following [8], where unsupervised methods generate node embeddings with GAT as the GNN-based encoder that are fed into a down-stream classifier as shown in Table S5, and semi-supervised learning following [14] with GCN, GIN and GAT as encoders as shown in Table S6, verifying the advantage of GraphCL.

**Table S5:** Comparing classification accuracy on top of learned node representations. The compared deep graph infomax (DGI, [8]) performance is from the original paper under the same experiment setting.

| Methods | Cora | Citeseer |
|---|---|---|
| DGI | 82.30±0.60 | 71.80±0.70 |
| NodeDrop v.s. Identical | 82.41±0.10 | 72.22±0.18 |
| NodeDrop v.s. NodeDrop | 81.76±0.17 | 73.14±0.15 |
| EdgePert v.s. Identical | 82.45±0.11 | 72.23±0.17 |
| EdgePert v.s. EdgePert | 82.32±0.15 | 73.11±0.19 |
| AttrMask v.s. Identical | 82.45±0.12 | 72.31±0.13 |
| AttrMask v.s. AttrMask | 81.78±0.17 | 72.05±0.22 |
| Subgraph v.s. Identical | 82.49±0.12 | 72.33±0.18 |
| Subgraph v.s. Subgraph | 81.71±0.14 | 73.12±0.17 |

**Table S6:** Node classification experiments with different models contrasting identical vs. different augmentation. Performance on the standard test sets of PATTERN SBM graphs. Results are averaged over 4 runs with 4 different seeds.

| Models | 10% baseline | NodeDrop | EdgePert | AttrMask | Subgraph |
|---|---|---|---|---|---|
| GCN | 70.53±0.72 | 67.91±0.38 | 68.61±0.69 | 68.11±0.51 | 67.79±0.33 |
| GIN | 96.61±2.77 | 97.52±0.57 | 98.09±0.49 | 97.23±0.42 | 98.35±0.49 |
| GAT | 76.71±8.27 | 75.86±3.36 | 79.73±7.08 | 84.72±2.00 | 85.30±2.31 |