[Reviews · NeurIPS 2020]

Review 1

Summary and Contributions: This paper proposes a contrastive learning algorithm to learn graph representations in an unsupervised manner. It is an extension of SimCLR [1] applied to learn graph representations that can be used for different graph classification tasks, either in semi-supervised learning, unsupervised learning or transfer learning scenarios. To do so, the authors propose several graph augmentation techniques that are needed for the contrastive learning algorithm, and analyse its effects on different types of datasets. The four different types of data augmentation techniques explored in the paper are: node dropping, edge perturbation, attribute masking and subgraph. In their empirical study, the authors explore the effect of these data augmentation techniques in different kinds of graph structure data like social networks and biochemical molecules, showing that different techniques work better on each domain, depending on the nature of the structure represented by the graph. This pre-training technique shows promising results across different datasets and tasks. [1] Chen, T., Kornblith, S., Norouzi, M., & Hinton, G. (2020). A simple framework for contrastive learning of visual representations. arXiv preprint arXiv:2002.05709.

Strengths: The main strength of this paper is the novelty of the proposed contrastive learning technique and the detailed experimental evaluation of it. Even though the proposed technique is adapted from a similar framework to learn representations using contrastive learning with visual data, the shift from images to graphs is not trivial, and most of the framework components need to be adapted to graph structured data. The empirical evaluation consists on a thorough analysis of the effect of different data augmentation techniques in the contrastive learning framework. The authors show that as expected, data augmentation is crucial for the proposed contrastive learning technique, but also that the data augmentation techniques to be used depend on the data’s domain, since the nature of the structure of the graph defines the meaning of each data augmentation technique.

Weaknesses: The technique presented in the paper would have been more complete if it had been evaluated for unsupervised node representation learning too. Node or edge level tasks are very relevant to graph structured data, and it would have increased the contribution and novelty of the paper. Additionally, some details are not entirely clear from the paper. For example, the different datasets used, or more details about the choice of encoder, readout and projection functions used.

Correctness: Yes

Clarity: Yes, the paper is generally well written.

Relation to Prior Work: It is discussed how the technique presented in this paper relates to previous work, specially SimCLR and Infograph, but the discussion could be longer and better structure. Also the comparison between this work and other unsupervised graph representation learning like Deep Graph Infomax would strengthen this section.

Reproducibility: Yes

Additional Feedback: The extension of this technique to node-level representation learning wouldn't be too different from the proposed technique, and it would strengthen the results of the paper if it showed improvement on node-level tasks as well. Regarding the Broader Impact statement, it reads like a conclusion or summary of the contributions of the paper to the graph representation learning field, instead of the original goal of this section. ====Post-rebuttal update===== The authors response what satisfactory.


Review 2

Summary and Contributions: In this paper, the authors propose GraphCL, a novel contrastive pre-training framework for graph representation learning. GraphCL first generates graph samples by applying four kinds of data argumentations on graphs, then it applies a contrastive loss to maximize agreement between graph embeddings of the same graph under different argumentations. The authors comprehensively studied four graph argumentations are proposed, namely node dropping, edge perturbation, attribute masking and subgraphs, through empirical experiments, and made some useful observations. Furthermore, experiments show that the pre-training technique proposed in GraphCL consistently benefits GNN performance and enable GNNs to be more robust to adversarial attacks.

Strengths: + Data augmentation on graphs is relatively under-explored in previous work. This paper empirically studies four kinds of graph augmentation techniques, making a novelty contribution to this field. + GraphCL conducts extensive experiments on both properties and influence of graph data argumentations and the performance of pre-training GNNs, and makes several useful observations. + The paper is clearly written and the idea is easy to follow.

Weaknesses: - The proposed GraphCL framework resembles contrastive framework in the vision domain such as SimCLR. The authors studied the role of data augmentations on graphs. Though extensive comparison have been made, the observations with respect to different augmentations on certain kinds of datasets are rather shallow, making this paper more like a survey paper. - No theoretical analysis is provided behind the proposed GraphCL framework. Why does GraphCL works by optimizing the contrastive objective? How does the loss differs from previous attempts, such as DeepWalk-like objectives? How do different data augmentations on graphs affect the mutual information between views? - GraphCL only considers pre-training the GNN at the graph level. However, node-level tasks are also very important. How does GraphCL differ from previous node-level work, such as DGI?

Correctness: Commonly-used protocols are applied in empirical evaluation. Findings are based on experimental results.

Clarity: The idea is easy to follow and the paper is clearly written.

Relation to Prior Work: There has been a lot of research which has leveraged contrastive learning: - Graph Representation Learning via Graphical Mutual Information Maximization, WWW 2020 - Unsupervised Attributed Multiplex Network Embedding, AAAI 2020 - Heterogeneous Deep Graph Infomax, arXiv preprint Moreover, the authors claim that none of existing work studied contrastive methods from the perspective of enforcing perturbation invariance (Line 96 -- Line 98). However, I in person believe DGI and InfoGraph, which construct negative (i.e. "corrupted") graphs by performing node-level permutation, do enforce perturbation invariance, as they still target at maximizing MI between input and embeddings. The main difference should be prior arts use different ways of constructing negative graphs.

Reproducibility: Yes

Additional Feedback: Please see my detailed comments. I would like to particularly hear from the authors regarding the following points. - I hope the authors can provide some insights into the motivation and methodology of the contrastive methods through deeper analysis on experimental results. - The authors are expected to give theoretical analysis on how different data augmentations on graphs affect the mutual information between views. ====Post-rebuttal comments===== I appreciate the authors' response to my concerns. However, I am still inlined to keep my score as I hope there will be in-depth discussion of the observations with different augmentations on certain kinds of datasets.


Review 3

Summary and Contributions: The paper proposed a method for pre-training graph neural networks. Its learning framework follows SimCLR, a SOTA method for pretraining CNNs. For data augmentation, authors design 4 methods to perturb graphs without affecting their semantic labels. Extensive experiments have been conducted to prove the effectiveness of the method. The paper is of good quality. My main concern is its novelty.

Strengths: - GNNs pre-training is a valuable topic to explore. - Authors transfer the contrastive learning framework to GNNs pretraining and prove its effectiveness through extensive experiments.

Weaknesses: - The contrastive learning framework is the same as SimCLR. - Graph augmentation methods, such as DropNode, DropEdge, FeatureMask, have been adopted in previous GNNs work, such as [1,2]. [1] DROPEDGE: TOWARDS DEEP GRAPH CONVOLUTIONAL NETWORKS ON NODE CLASSIFICATION. [2] STRATEGIES FOR PRE-TRAINING GRAPH NEURAL NETWORKS.

Correctness: To the best of my knowledge, the method is correct.

Clarity: The paper is well written and clearly organized.

Relation to Prior Work: Yes.

Reproducibility: Yes

Additional Feedback:


Review 4

Summary and Contributions: The authors propose a graph contrastive learning (GraphCL) framework to learn perturbation-invariant unsupervised representations of graph data. The method can produce graph representations of similar or better generalizability, transferability, and robustness compared to state-of-the-art methods. They also proposed several graph data augmentations.

Strengths: 1) The paper is well-organized with motivation and approaches clearly stated and illustrated. 2) The experiments results verify the state-of-the-art performance of the proposed framework in both generalizability and robustness 3) The proposed GraphCL framework is novel and worth further exploring.

Weaknesses: 1) Missing Citations and prior work: [1] The MOCO paper is missing: Kaiming He, et al., Momentum Contrast for Unsupervised Visual Representation Learning, CVPR 2020. [2] Herzig, et al., Learning Canonical Representations for Scene Graph to Image Generation, ECCV 2020. Recently, the authors showed in [2] that by using a canonicalization process for graphs, the information is propagated in the graph better than deeper networks, and thus they can generate complex visual scenes. Maybe it could be relevant as a new data augmentation (see next bullet) 2) The data augmentations are very straightforward. I wonder if more sophisticated augmentations could be used, such as capturing invariance to logical equivalences (see [2]), permutation invariant for the nodes, and more. 3) Although the graph domain is new, the graph contrastive learning framework seems to be employed from past works (SimCLR and MOCO).

Correctness: Yes

Clarity: Overall this paper is well written and the technical details are easy to follow.

Relation to Prior Work: Yes.

Reproducibility: Yes

Additional Feedback: ================ POST-REBUTTAL UPDATE: ======================== After reading the authors' feedback and other reviewers' opinions, I would like to thank the authors for their rebuttal. The rebuttal addresses my concerns. I believe this paper contains an interesting novelty, which is a promising direction worth investigating, and thus I would like to accept the paper. I raised my score to 7. ========================================================================

[Author Response · NeurIPS 2020]

We gratefully appreciate all reviewers' suggestions. Below please find the responses to the specific comments.

▷ **General responses. Q: Existing contrastive learning (CL) framework & graph data augmentation methods.**
*Reply:* As reviewers #1&2 point out, (i) it is nontrivial to develop the CL framework that shifts from images to graphs,
since most components need to be adapted; and (ii) though present in previous work, data augmentation for graphs
is under-explored, whereas our thorough experiments revealed insights into the role of augmentation combinations
in GraphCL for various graph data. Furthermore, GraphCL generalizes previous works in what and how structural
invariance in graphs is captured, which will be detailed in responses to reviewer #2. **Q: Theoretical analysis.** *Reply:*
(i) A major stream of theoretical analysis for CL is currently based on mutual information (MI). For instance, we can
view the contrastive objective in GraphCL as maximizing the MI between graph views by generalizing the Deep Graph
Infomax (DGI) approach (see $\mathbf{Q}^*$ of reviewer #2), which can be included in the final version for completeness. (ii)
However, we also emphasize that the MI-view of CL has been challenged recently and more theoretical development
is needed. For instance, [1] shows that the success of the Infomax approaches is only loosely connected to MI but
strongly depends on the bias in encoders & estimators. We instead may explore some newer perspectives (e.g. [2]) in
the future work. We believe that our work provides comprehensive guide to the practice of graph representation learning
and inspirations for new theoretical development. **Q: Node-level representation learning and tasks.** *Reply:* We
implemented GraphCL for a node-level task based on the experiment setting in DGI [3], where Subgraph and NodeDrop
are respectively chosen for Cora and Citeseer based on validation. The preliminary results below are promising and
more tasks/datasets will be included during revision.

| Dataset | DGI | GraphCL | Dataset | DGI | GraphCL |
|---------|-----|---------|---------|-----|---------|
| Cora | 82.3% | 82.4% | Citeseer | 71.8% | 73.1% |

▷ **Reviewer #1. Q: Strengthening the writing of related work, experiment settings and broader impact.** *Reply:*
Thanks for detailed comments. We will revise to improve the contents and the structure as suggested.

▷ **Reviewer #2. $\mathbf{Q}^*$: Insights into the GraphCL motivation and methodology** *Reply:* Recalling the contrastive
objective (Eq. (3) in the paper) in GraphCL, we can rewrite the loss function in the expectation form for each batch as:

$$l = -\mathbb{E}_{\mathbb{P}_{(\hat{\mathbb{G}}_i, \hat{\mathbb{G}}_j)}} \{ \text{sim}(g(f(\hat{\mathcal{G}}_i)), g(f(\hat{\mathcal{G}}_j)))/\tau \} + \mathbb{E}_{\mathbb{P}_{\hat{\mathbb{G}}_i} \times \mathbb{P}_{\hat{\mathbb{G}}_j}} \{ \log \sum_{\hat{\mathcal{G}}_j} \exp(\text{sim}(g(f(\hat{\mathcal{G}}_i)), g(f(\hat{\mathcal{G}}_j)))/\tau) \}$$

$$= -\mathbb{E}_{\mathbb{P}_{\hat{\mathbb{G}}_i}} \{ \mathbb{E}_{\mathbb{P}_{(\hat{\mathbb{G}}_j | \hat{\mathbb{G}}_i)}} T_w(f(\hat{\mathcal{G}}_i), f(\hat{\mathcal{G}}_j)) - \mathbb{E}_{\mathbb{P}_{\hat{\mathbb{G}}_j}} \log \sum_{\hat{\mathcal{G}}_j} \exp(T_w(f(\hat{\mathcal{G}}_i), f(\hat{\mathcal{G}}_j))) \}, \tag{1}$$

where $\mathbb{P}_{(\hat{\mathbb{G}}_i, \hat{\mathbb{G}}_j)}, \mathbb{P}_{(\hat{\mathbb{G}}_i | \hat{\mathbb{G}}_i)}, \mathbb{P}_{\hat{\mathbb{G}}_i}$ are respectively the joint, conditional and marginal distribution, and $T_w : \mathbb{R}^D \times \mathbb{R}^D \to \mathbb{R}$
is a learnable score function that we parametrize with the similarity function $\text{sim}(\cdot, \cdot)$ and the projection head $g(\cdot)$.
Notice that Eq. (1) fits the formulation of the InfoNCE loss [4, 2] and therefore minimizing the contrastive objective
(1) is explicitly maximizing a lower bound of the mutual information between the latent representations of two views
of graphs $f(\hat{\mathcal{G}}_i), f(\hat{\mathcal{G}}_j)$. Therefore, we treat our contrastive objective (Eq. (1)) as a general formulation for graph
contrastive learning, that can be instantiated as a specific algorithm by designing how to construct different views of
graphs, where augmentation is adopted as a general method for view constructions. **Q: Related work.** *Reply:* We will
include and discuss the suggested papers in revision. We emphasize that our GraphCL framework generalizes previous
works for graphs in what to contrast (various augmentation options, unified as a general method for view constructions)
and how to contrast (contrastive objective (Eq. (1)) as a general formulation). For instance, the Infomax approaches
[3] only assume fixed views of original graphs and subgraphs in the objective (1) for approximation. Thus, previous
works can be regarded as special cases of GraphCL that is a more general framework to learn structural invariance from
diverse graph data without making specialized assumptions.

▷ **Reviewer #3.** Please refer to the general responses for the answers.

▷ **Reviewer #4. Q: Missing citations and prior work** *Reply:* Thanks for bringing some missing related work to our
attention. We will add them during revision. **Q: More sophisticated graph data augmentation.** *Reply:* Thanks for
bringing [5] to our attention for designing potential more complicated augmentation. Although the ECCV acceptance is
after the NeurIPS submission, we find the paper interesting proposing domain specific augmentation (for scene graph in
computer vision), and will definitely explore it in our final version.

[1] Michael Tschannen, et al. On Mutual Information Maximization for Representation Learning. ICLR, 2020.
[2] Yonglong Tian, et al. What Makes for Good Views for Contrastive Learning. ECCV, 2020.
[3] Petar Veličković, et al. Deep Graph Infomax. ICLR, 2019.
[4] Aaron van den Oord, et al. Representation Learning with Contrastive Predictive Coding. arXiv, 2018.
[5] Roei Herzig, et al., Learning Canonical Representations for Scene Graph to Image Generation. ECCV, 2020.


[Meta-Review · NeurIPS 2020]

The author proposes a contrastive learning framework for graph embedding. It first generates graph samples by applying several graph augmentation strategies (node trapping, edge perturbation, attribute masking and sub-graphing) to the original graph, and then maximize the agreements between the graph embeddings of the same graph under different argumentations. In other words, it aims to learn the perturbation-invariant embedding of graphs. Pro: 1. Although contrastive learning via data argumentation has been studied with other data types (e.g., images), adapting such idea to graphs-structured data is non-trivial and hence a novel contribution in this paper. 2. The empirical evaluation is extensive, and the analysis is insightful. 3. The paper is well written and easy to follow Cons: 1. A comparison of node- or edge-level embedding methods would increase the completeness of experiments. 2. More in-depth (theoretical) analysis/comparison of different argumentation strategies and contrastive losses would be a plus. 3. Two citations on contrastive learning framework are missing [1][2]. [1] Kaiming He, et al., Momentum Contrast for Unsupervised Visual Representation Learning, CVPR 2020. [2] Herzig, et al., Learning Canonical Representations for Scene Graph to Image Generation, ECCV 2020.